# Discovery and Validation of Methylation Signatures in Circulating Cell-Free DNA for the Detection of Colorectal Cancer

**DOI:** 10.3390/biom14080996

**Published:** 2024-08-13

**Authors:** Zhiping Long, Yu Gao, Zhen Han, Heli Yuan, Yue Yu, Bing Pei, Yanjie Jia, Jingyu Ye, Ying Shi, Min Zhang, Yashuang Zhao, Di Wu, Fan Wang

**Affiliations:** 1Department of Epidemiology, School of Public Health, Harbin Medical University, Harbin 150028, China; 202101040@hrbmu.edu.cn (Z.L.); gaoyu@hrbmu.edu.cn (Y.G.); 2020020160@hrbmu.edu.cn (Z.H.); 2022020094@hrbmu.edu.cn (H.Y.); 2020020138@hrbmu.edu.cn (Y.Y.); 202301087@hrbmu.edu.cn (B.P.); 2022020085@hrbmu.edu.cn (Y.J.); 2021020292@hrbmu.edu.cn (J.Y.); 2023020198@hrbmu.edu.cn (Y.S.); 2022020101@hrbmu.edu.cn (M.Z.);; 2Department of Colorectal Surgery, Tumor Hospital of Harbin Medical University, Harbin Medical University, Harbin 150081, China

**Keywords:** colorectal cancer, methylation, WGBS, cfDNA, biomarkers

## Abstract

This study was conducted with the primary objective of assessing the performance of cfDNA methylation in the detection of colorectal cancer (CRC). Five tumor tissue, 20 peripheral blood leucocyte, and 169 cfDNA samples were collected for whole-genome bisulfite sequencing (WGBS) analysis. Bioinformatic analysis was conducted to identify differentially methylated regions (DMRs) and their functional characteristics. Quantitative methylation-specific PCR (qMSP) was used to validate the methylation levels of DMRs in the tissues and leucocytes. cfDNA samples from CRC patients and healthy controls were used to evaluate the performance of the DMR analysis. WGBS analysis revealed a decrease in DNA methylation levels in the CpG context in CRC tumor tissues compared with adjacent normal tissues. A total of 132 DMRs in cfDNA were identified as potential markers for diagnosing CRC. In a cohort of 95 CRC patients and 74 healthy controls, a combination of the three DMRs (*DAB1*, *PPP2R5C*, and *FAM19A5*) yielded an AUC of 0.763, achieving 64.21% sensitivity and 78.38% specificity in discriminating CRC patients from healthy controls. This study provides insights into DNA methylation patterns in CRC and identifies a set of DMRs in cfDNA with potential diagnostic value for CRC. These DMRs hold promise as biomarkers for CRC detection, offering promise for non-invasive CRC diagnosis. Further research is warranted to validate these findings in larger cohorts.

## 1. Introduction

Colorectal cancer (CRC) is one of the deadliest cancers with a poor prognosis and increasing incidence worldwide [1]. Numerous studies underscore the critical impact of early-stage diagnosis on CRC survival, with later-stage diagnoses exhibiting diminished survival rates [2,3,4]. Consequently, the timely detection of early-stage CRC, amenable to curative interventions, is imperative for prolonging survival. Despite the recommendation of colonoscopy for high-risk individuals, its invasive nature and high costs present substantial barriers to widespread participation [5]. Non-invasive screening methods, such as stool blood-based or DNA-based tests like the fecal immunochemical test for hemoglobin (FIT), aim to enhance participation rates but fall short in reliably detecting early-stage CRC and precursor lesions (adenomas) [6]. Peripheral blood-based testing offers superior advantages in terms of patient compliance [7]. However, widely used plasma biomarkers like CA19-9 and CEA exhibit inadequate sensitivity and specificity, rendering them impractical for CRC screening and diagnosis [8].

Achieving early detection at the population level necessitates not only optimal specificity and sensitivity but also precise cancer origin localization [9,10]. Among liquid biopsy tests, identifying genomic and epigenomic alterations in circulating cell-free DNA (cfDNA) has emerged as a promising approach. Advances in next-generation sequencing of cfDNA alterations, exemplified by CancerSEEK and PanSeer, demonstrate impressive preliminary screening performance [11,12]. In comparison to genomic alterations such as mutations and copy number variations, cfDNA methylation alterations manifest earlier with abundant cancer- and origin-specific signals [13,14]. Thus, they hold promise for effective non-invasive early CRC detection [15,16,17].

However, few potential blood biomarkers are currently being used in routine screening and diagnosis in clinical practice. Our previous study has already demonstrated the usefulness of cfDNA differentially methylated position (DMP) markers of CRC, which have shown excellent diagnostic performance [18]. The statistical association between methylation levels and phenotype has been identified at the level of individual CpG sites (DMPs) and broader genomic regions. Hence, the screening and validation of methylation markers at a regional level, as opposed to a single CpG level, holds the potential to significantly enhance the tumor specificity of these markers. Compared with DMPs, differentially methylated region (DMRs) may be more biologically relevant and are more likely to be associated with modified gene expression because of the strong correlation among adjacent CpGs [19]. 

In this study, we aimed to identify CRC-specific DMRs by whole-genome bisulfite sequencing (WGBS) and to evaluate the potential performance of cfDNA methylation markers in the detection of CRC. We first compared the methylation profiles between CRC tumor and paired adjacent tissues from the WGBS data and identified CRC-specific DMRs. Then, considering the potential noise in the DNA methylation of peripheral blood leukocyte samples, only the DMRs detected in cfDNA samples were selected as a potential diagnostic biomarker set. Finally, the diagnostic performance of the three DMRs on the *DAB1*, *PPP2R5C*, and *FAM19A5* genes within the cfDNA DMR set was validated in an independent validation cohort.

## 2. Materials and Methods

### 2.1. Patient Sample Acquisition

Blood and tissue samples from all participants were procured at the Second Affiliated Hospital and Tumor Hospital of Harbin Medical University. The schematic representation and methodologies employed for methylation analysis, as well as the validated cfDNA targets in CRC, are elucidated in Appendix A. In the preliminary discovery cohort, adjacent and tumor tissues were collected from 5 colorectal cancer patients. Additionally, 10 mL of peripheral blood was drawn for the extraction of peripheral blood leukocytes and cfDNA. The in-house primary validation cohort included 20 paired colorectal cancer tissue samples and corresponding adjacent tissues, along with their respective peripheral blood leukocyte samples. The in-house cfDNA validation cohort encompassed 169 cfDNA samples from both colorectal cancer patients and healthy controls. Healthy individuals, matched with the case group in terms of age and sex and without any history of tumors or colorectal diseases, were recruited as the control group. All patient cfDNA samples were acquired between February 2019 and December 2021. Demographic and clinical information were meticulously retrieved from medical records and questionnaires, with informed consent obtained from all subjects. The study protocol received approval from the Medical Ethics Committee of Harbin Medical University. Comprehensive clinical characteristics for all study participants are detailed in Appendix A.

### 2.2. Whole-Genome Bisulfite Sequencing and Targeted Methylation Quantification

Genome-wide bisulfite sequencing (WGBS) was performed using the Illumina HiSeq 4000 platform (Illumina, San Diego, CA, USA) to evaluate DNA methylation in 5 pairs of tumors and adjacent tissues, as well as from peripheral blood leucocytes and cfDNA samples according to standard procedures. Several bioinformatic steps were carried out to analyze the bisulfite sequencing reads: (1) adapter trimming, (2) alignment of the bisulfite-treated reads to the human genome, (3) determination of the methylation state at each cytosine, and (4) filtering of the contaminating reads. Then, the callDMR function in the DSS software was utilized to detect DMRs (the parameters: delta = 0.1, p.threshold = 0.05) [20]. Finally, the specific DMRs selected were further validated by quantitative methylation-specific PCR (qMSP) and droplet digital PCR (ddPCR) in the tissue, leukocyte, and cfDNA samples. More details about the high-throughput sequencing, bioinformatic preprocessing and targeted methylation quantification can be found in the Appendix A.

### 2.3. Statistical Analysis

Statistical analysis was performed using the R software (R software, version 4.1.0). Diagnostic accuracy was described by using the means of ROC analyses. AUCs were reported, including the 95% confidence intervals (CIs). The Wilcoxon rank–sum test was used for the comparison of continuous variables between the two groups and conducted on a two-sided basis after the equality of variances had been assessed with the Levene test; otherwise, the Kruskal-Wallis test was used for comparison. The correlation analysis was performed using Spearman’s rank correlation coefficient. All studies were statistically significant at *p* < 0.05. Visualization of the gene functional enrichment analyses were performed by the “clusterprofile” R package.

## 3. Results

### 3.1. DNA Methylation Atlas of CRC Sample Screening and Evaluation

In this study, a meticulous scrutiny of the samples was undertaken, focusing on the bisulfite conversion rate, coverage, and Q20 rate subsequent to a comprehensive WGBS analysis of all 12 samples. The sample set comprised five pairs of tumor and adjacent tissues, one mixed sample of peripheral blood leukocytes, and one mixed sample of cfDNA. All samples met the necessary criteria for subsequent analysis. Clean reading and mapping rates were determined for each sample post-filtering. The bisulfite conversion rate, a key indicator of the methylation-level detection reliability, consistently exceeded 99%, affirming the robustness of the calculated CpG-site methylation levels. The single-base resolution methylation ratio exhibited an average coverage of 80.64% (±2.01% standard deviation (SD)). Quality evaluation indicated Q20% rates above 96% and Q30% rates above 89%, meeting acceptable standards (Table 1). A Circos plot was used to visually represent DNA methylation levels in various sequence contexts (mCG, mCHG, and mCHH) on human chromosomes (1–22 chromosome; Appendix A).

### 3.2. Global DNA Methylation Patterns in CRC Tissues

Methylation levels were calculated using a 2 Kb/bin sequencing environment, and Pearson’s correlation coefficient analysis was performed, confirming the high consistency and robust repeatability of our samples (Figure 1A). To discern global DNA methylation profile differences between tumor and adjacent tissues, we analyzed DNA methylation levels in three contexts: CG, CHG, and CHH (where H is A, C, or T). Figure 1B illustrates that the majority (94%) of cytosines were methylated in the CpG context, with a minimal proportion (5%) methylated in non-CG contexts (CHG and CHH). Exploring methylated cytosine patterns in tumor tissues, we analyzed genome-wide mC sequence preferences in various contexts. Our results highlighted a preference for methylated cytosines to be located within CG, CHG, and CHH (H = A > T) contexts (Figure 1C). Consistent with expectations, a genome-wide decrease in DNA methylation levels in the CpG context was observed when comparing tumor tissues with adjacent normal tissues (*p* < 0.001) (Figure 2A).

### 3.3. Enrichment Analysis and Functional Characterization of DMGs

A total of 12,923 DMRs were identified in the CpG context, with 62.82% hypermethylated and 37.18% hypomethylated. Notably, a substantial proportion of DMRs were located in CGI, CGI_shore, promoter, intron, and exon regions (Figure 2B). DMR lengths predominantly fell within the range of 100–200 bp, with all being less than 1000 bp (Figure 2C). Through gene annotation of the DMRs, we identified numerous hypermethylated and hypomethylated genes. To explore the potential biological roles of differentially methylated genes (DMGs), gene ontology (GO) analysis and KEGG pathway analysis were conducted. Our findings revealed that DMGs were primarily enriched in tumor-invasion-related axonogenesis, cell adhesion (biological processes and cell components), and cancer-related transcription factor regulation (molecular functions) (Appendix A), as well as the PI3K-Akt, Rap1, MAPK, and Ras signaling pathways (Appendix A).

### 3.4. Discovery of Universal CRC-Specific Methylation Markers

Our primary aim was the identification of liquid-biopsy-appropriate biomarkers for CRC. This involved a comprehensive analysis of WGBS data from cfDNA and peripheral blood leukocyte samples extracted from colorectal cancer patients. In peripheral blood leukocytes, 40,286,113 methylated sites were identified, with 71,660 located in DMRs that differentiated colorectal cancer tissue from adjacent tissue. Among these DMRs, 3933 exhibited heightened methylation levels in tumor tissue. For cfDNA, 1,548,481 methylated sites were identified, with 1251 in DMRs that differed between colorectal cancer tissue and adjacent tissue. Within these DMRs, 432 displayed increased methylation in tumor tissues. Additionally, we scrutinized the methylation level distribution across the chromosomes and genomic functional elements in colorectal cancer tissue, peripheral blood leukocytes, and cfDNA. The results revealed a largely consistent average methylation level distribution among the three entities (Figure 3; Appendix A). Utilizing Venny 2.1 software, we isolated unique targets within the cfDNA, identifying 132 exclusively highly methylated DMRs in cfDNA (Figure 4A, Appendix A). 

### 3.5. Evaluation of Candidate Methylation Biomarkers in Tissue and Leukocyte Samples

In this step, we gathered 20 paired CRC tumor tissue and adjacent normal tissue samples to validate the methylation level differences within the identified DMRs. Demographic characteristics of the 12 males and 8 females are detailed in Appendix A, with an average age of (59.05 ± 12.57) years. Initially, we identified 132 DMRs with significant potential as biomarkers. We ranked these DMRs based on the magnitude of the methylation differences, prioritizing those with the largest differences between CRC patients and healthy controls. During this process, we excluded genes that had already been extensively studied by others, as well as those that did not show consistent results in our preliminary experiments with blood samples. This thorough screening process led us to focus on the three genes (*DAB1*, *PPP2R5C*, and *FAM19A5*) that demonstrated the most promising and consistent performance. Figure 4B, Appendix A, and Appendix A highlight significant methylation level differences in DMRs within the *DAB1*, *PPP2R5C*, and *FAM19A5* genes between the 20 pairs of tumor and adjacent normal tissues. Furthermore, we extended our analysis to assess methylation within the identified DMRs in DNA samples from peripheral blood leukocytes collected from the same 20 CRC cases. Appendix A illustrates the melting curve analysis conducted on the peripheral blood leukocyte samples; no amplification of the target product was detected, signifying the absence of methylation in the target DMRs within the peripheral blood leukocyte samples. Additionally, we also performed a comparison of the qMSP and ddPCR methods for their detection sensitivity and specificity when using cfDNA. Detailed results can be found in the Appendix A. The comparison revealed that qMSP demonstrated a performance comparable to that of ddPCR in detecting DMR methylation in cfDNA, with a detection limit as low as 100 copies, effectively distinguishing non-specific amplification.

### 3.6. cfDNA-Based Validation in CRC and Healthy Controls

A cohort of 95 CRC patients and 74 healthy individuals were enrolled for this study. Comprehensive participant characteristics are summarized in Appendix A. Colorectal cancer patients had a mean age of (62.53 ± 10.98) years, while the control group’s mean age was (60.46 ± 12.49) years. No statistically significant age difference was observed between the two groups, particularly when stratified by age above 60 years (*p* > 0.05). Within the colorectal cancer group, there were 54 males (56.8%) and 41 females (43.2%), while the non-cancer control group comprised 24 males (33.3%) and 48 females (66.7%). A notable statistical difference was observed in the gender distribution between the two groups (*p* = 0.003). The area under the curve (AUC) values for the three DMRs in distinguishing CRC patients from healthy controls were as follows: 0.688 (95% CI: 0.620–0.756) for *DAB1*, 0.647 (95% CI: 0.588–0.707) for *PPP2R5C*, and 0.639 (95% CI: 0.577–0.701) for *FAM19A5*. The combined analysis of all three DMRs yielded an AUC of 0.763 (95% CI: 0.696–0.830), discriminating CRC patients from healthy controls with 64.21% sensitivity and 78.38% specificity (Figure 4C; Appendix A).

### 3.7. Relationship between DMR Methylation Levels of cfDNA and Clinicopathological Features

To explore the correlation between the methylation status of the DMR on the *DAB1* gene and clinicopathological features of colorectal cancer, patients were categorized into methylated and unmethylated groups. The results, presented in Appendix A, revealed that 11 patients (29.7%) in the *DAB1* methylation group exhibited vascular tumor invasion, compared with 17 patients (60.7%) in the non-methylation group. Significantly, the presence of vascular tumor invasion was associated with *DAB1* gene methylation (No vs. Yes; OR = 0.274, 95% CI = 0.097–0.771) (*p* = 0.014). No other clinicopathological characteristics demonstrated a significant association with methylation of the DMR in the *DAB1* gene. Additionally, regarding the DMRs of the *PPP2R5C* and *FAM19A5* genes, no significant correlations were found between the pathological characteristics and the methylation levels of these DMRs (*p* > 0.05).

## 4. Discussion

Epigenetic abnormalities can lead to the development of tumors; in particular, aberrant DNA methylation patterns are associated with the diagnosis and prognosis of many types of cancer [21]. This study aims to identify potential CRC diagnostic biomarkers based on cfDNA methylation. In this research, we examined the genome-wide methylation landscape of primary CRC tumors and compared it with the methylation of cfDNA and white blood cells to identify potential CRC diagnostic biomarkers. Through the analysis of WGBS data from these three types of samples, we discovered a significant number of highly methylated DMRs in CRC tumor tissues, and notably, the hypermethylation status of specific DMRs detected exclusively in cfDNA samples can serve as an effective approach for CRC diagnosis.

### 4.1. Application of WGBS in Screening for CRC Methylation Biomarkers

DNA methylation is one of the earliest molecular alterations occurring in the process of tumorigenesis [22]. Currently, the most widely used platform for discovering cfDNA methylation biomarkers in the exploration phase is the Infinium HumanMethylation450 BeadChip array, also known as the “450K array” [23]. But the limitations of the 450K array are its coverage of only approximately 2% of all CpG sites in the genome, with a significant portion of genomic information missing. Thus, the majority of studies on cfDNA methylation have focused on the characterization of single CpG sites. However, it has been recognized that concomitant changes spanning entire genomic regions, referred to as DMRs, are common in cancer tissues in comparison with normal tissues [24]. Therefore, accurate DMR identification is critical to enable a thorough understanding of the extent of localized differential methylation in relation to the phenotype of CRC. WGBS is the most comprehensive and non-biased methylome profiling method, since it can cover the highest proportion of the genome among all DNA methylation detection methods, and it is very appropriate for studies that are interested in the methylation profiles of not only genic regions but of broader intergenic regions [25]. However, its extensive sequencing costs currently restrict its widespread application. As far as we know, there has been no research exploring the use of WGBS datasets to identify DMRs as diagnostic biomarkers for CRC. 

Our analysis of WGBS data from tissue samples of colorectal cancer patients indicates that aberrant DNA methylation is a common feature in CRC. The DNA methylation levels in tumor tissues exhibit a genome-wide decrease, consistent with previous reports of genome-wide hypomethylation in various human cancers [26,27,28]. Differential analysis has identified over 10,000 DMRs, signifying regions that manifest methylation alterations between CRC tumor tissues and matched distant normal tissues. These regions are not only confined to known promoters and CpG island areas but also encompass exonic and intronic regions. Typically, methylation of promoter CpG islands displays a negative correlation with the expression of the same gene. Given that the majority of our DMRs (62.82%) are highly methylated, this suggests that these methylation changes may have a role in gene regulation. In contrast, a study based on reduced-representation bisulfite sequencing (RRBS) reported an overall reduction in methylation in primary colon tumors compared with normal adjacent tissues [29]. In that study, the majority of the identified DMRs (75%) in stage III–IV CRC tumor tissues exhibited hypomethylation. This disparity is likely, at least in part, attributed to the utilization of different methylation sequencing technologies in various studies since they may cover distinct genomic regions. In our study, we employed WGBS, enabling us to cover a larger proportion of the genome, particularly non-CGI (CpG island) regions. We observed extensive hypermethylation in regions characterized by low CpG density. In summary, our study, employing WGBS, provides a comprehensive view of CRC methylation alterations, emphasizing the heterogeneity of hypermethylation and hypomethylation across diverse genomic regions.

Furthermore, within the DMGs, the results of the enrichment analysis highlight a significant correlation between DMRs and the malignant phenotype of tumors. Additionally, some of these genes are characterized by known functions. For instance, GAGE12E, a crucial member of the cancer–testis antigen family, is associated with tumor immunity [30]. MEX3D, a Mex-3 RNA-binding family member, plays a role in the pathological and physiological processes of tumor differentiation, proliferation, autophagy, apoptosis, and inflammation [31]. The DUOX2 gene is a glycoprotein and a member of the NADPH oxidase family, with connections to colorectal reactive oxygen species (ROS), inflammation, and microbial dysbiosis [32]. However, the majority of these DMGs do not exhibit a clear relationship with the onset and development of cancers, including CRC. On the other hand, the mechanisms by which the methylation of specific DMRs impacts the expression of their associated genes are exceedingly intricate. Some transcription factors can even preferentially recognize methylated CpGs and activate over a hundred genes [33]. Further exploration of the potential functional mechanisms of these DMRs may deepen our understanding of colorectal cancer’s origins and offer potential therapeutic targets.

### 4.2. Research Progress on CRC cfDNA Methylation Diagnostic Biomarkers

To robustly screen a set of potential biomarkers, the majority of current studies employ the 450K array for the analysis of several hundred samples [34]. This approach has identified a plethora of DMPs for early diagnosis or CRC risk assessment, with these DMPs utilizing DNA from tissues, blood, feces, urine, and colonic lavage fluid [23]. For instance, the THUNDER study developed and validated the MCDBT-1 and MCDBT-2 models using a customized panel of 161,984 CpG sites to analyze cfDNA from 1693 participants [9]. These models demonstrated impressive performances in detecting six types of cancers (colorectal, esophageal, liver, lung, ovarian, and pancreatic), with MCDBT-1 achieving a sensitivity of 69.1% and a specificity of 98.9% in an independent validation cohort. For early-stage cancers (I-III), MCDBT-1 had a sensitivity of 59.8%. The MCDBT-2 model, while having slightly lower specificity at 95.1%, offered a higher sensitivity at 75.1%, demonstrating the model’s potential for improving early cancer detection and patient outcomes. However, being limited by current screening and diagnostic techniques, it is difficult to translate these multi-CpG site diagnostic models into practical applications. Notably, CpG methylation in the genome often occurs in clusters, and CpGs in the same region are correlated and perform a similar function. Identification of these correlated regions not only reduces the data dimensions but also increases the detection power by exploiting nearby CpG information [25].

Throughout our exploration, we employed a novel strategy utilizing WGBS to systematically screen for differentially methylated regions within the colorectal cancer genome, identifying potential novel DNA methylation biomarkers with enhanced effectiveness. In cancer patients, tumor cells are not the only producers of cfDNA as other non-cancer cells also playing a crucial role in its release, with the hematopoietic system being the primary source of cfDNA in healthy subjects. The release of cfDNA from non-cancer cells can lead to distortions in cancer cell signals [35]. Therefore, we applied single-base-resolution methylation sequencing to tissue, blood leukocyte, and cfDNA samples from CRC patients, enabling the identification of specific DNA methylation changes (DMRs) in the CRC genome within cfDNA by eliminating interference from leukocyte-released cfDNA. Given the relatively small sequencing sample sizes, these DMRs were further validated in tumor tissue and leukocyte samples using quantitative methylation-specific qPCR (qMSP). Our findings strongly suggest that using cfDNA fragmentation patterns, which depend on DMRs, could serve as a promising alternative method for diagnosing CRC. 

Moreover, the qMSP for cfDNA methylation exhibited concordance with the outcomes obtained through digital droplet PCR (ddPCR). In contrast, ddPCR is preferred over traditional PCR for cfDNA detection and possesses great sensitivity and specificity. However, its workflow is complex, and it is costly for typical clinical work [36]. The SYBR Green-based qMSP method offers several advantages, including cost-effectiveness and the ability to perform melting curve analysis to confirm the specificity of the amplified product. By eliminating the need for fluorescently-labeled probes, this method reduces the overall cost of the assay and simplifies the procedure. However, it is important to note that while SYBR Green-based methods provide a cost-effective and straightforward approach, specific probes can offer higher specificity in detecting target sequences. Despite this, the simplicity and requirement for commonly used equipment in SYBR Green-based qMSP make it a promising option for enhancing the integration of cfDNA methylation analysis into clinical practice.

In this study, we have identified three DMRs located within the *DAB1* promoter region, *FAM19A5* promoter region, and *PPP2R5C* intronic region that exhibit higher methylation levels in cancer tissues compared with adjacent tissues in Chinese colorectal cancer patients. Specifically, the promoter region of *FAM19A5* in CRC tumor tissues from the TCGA dataset also displayed elevated methylation levels compared with adjacent normal tissues [37]. Therefore, the DNA methylation biomarkers discovered in our study may potentially enhance the effectiveness of biomarker-based CRC detection in Asian populations. In addition, our analysis did not reveal significant correlations between the methylation levels of these DMRs and TNM staging of the CRC. This lack of correlation could be attributed to several factors. Firstly, the methylation changes we identified may occur early in the carcinogenesis process and might not be directly linked to the tumor’s progression or stage. This suggests that these DMRs could serve as early detection markers rather than indicators of tumor stage. Additionally, the heterogeneity of CRC could mean that methylation changes vary widely among individuals and may not align consistently with TNM staging. Moreover, we acknowledge that while our cohort included CRC patients of various stages, the sample size for each stage may not have been sufficiently large to detect subtle stage-specific methylation changes. Future studies with larger cohorts, stratified by stage, are necessary to fully explore the potential correlations between these DMRs and tumor staging.

The FDA has approved SEPT9 methylation in cfDNA as the sole blood-based biomarker for detecting CRC [38]. Other defined markers for CRC detection described in more than one study include methylated *BCAT1, IKZF1, SDC2, ALX4, SFRP2, OSMR, SFRP1*, and *VIM* [39]. The combined study of *ALX4, BMP3, NPTX2, RARB, SDC2, SEPT9*, and *VIM* showed a sensitivity of 88.7% for stages I/II and 90.7% for all CRC stages using a model that took into account the variables of female gender and age above 66 years, with a *p*-value > 0.05 [40]. Notably, most screening populations in these studies come from the TCGA database, where over 75% are of Caucasian descent. In Caucasians, SEPT9 exhibits high levels of methylation in CRC, achieving an AUC of 0.8 for distinguishing CRC from healthy controls using blood mSEPT9 [41]. However, the reported diagnostic sensitivity for blood-based mSEPT9 varies [42], with Lee et al. finding low sensitivity (36.6%) in the Korean population [43]. Church et al. conducted a large-scale trial with 7941 patients, showing relatively low sensitivity (48.2%) and high specificity (91.5%) for circulating methylated-SEPT9 DNA testing [44]. Therefore, when studying cfDNA methylation biomarkers, it is crucial to consider individual variations in DNA methylation profiles across ethnicities [45], which may influence the detection performance of cfDNA methylation.

Currently, there is a lack of research on the relationship between DNA methylation levels of the *DAB1*, *FAM19A5*, and *PPP2R5C* genes and colorectal cancer. Some research findings suggest abnormal methylation of *PPP2R5C* in colorectal cancer tumor tissues and fecal DNA. In 2021, Cheng et al. [46]. identified a set of biomarker genes (*ADHFE1*, *SDC2*, and *PPP2R5C*) for CRC detection through the analysis of TCGA 450K methylation data and Taiwan NHIRD EMR data. The potential disease relevance of these selected genes was validated using methylation testing based on fecal DNA. The integration of methylation testing for these three genes achieved a sensitivity and specificity of 84.6% and 92.3% for CRC detection. DNA methylation detection can be performed using both blood and fecal samples. Both of these approaches hold the potential to enhance the screening and diagnostic compliance among the general population for CRC. Blood-based DNA methylation testing is considered to have better compliance. It is estimated that in China, fewer than 20% of eligible individuals have undergone a colonoscopy, primarily due to its inconvenience and invasiveness [47]. The cfDNA methylation markers discovered in this study offer a potential foundation for the development of novel non-invasive and effective diagnostic tools, which could improve the compliance in CRC detection.

While the findings of this study still require validation in larger datasets, the colorectal-tissue-specific methylation markers obtained through high-throughput whole-genome methylation data screening represent the current mainstream and cost-effective approach for marker selection. This approach mitigates the potential confounding effects introduced by peripheral blood leukocyte DNA methylation. In the marker validation phase, qMSP, a highly sensitive and operationally feasible technique, is employed to quantify the methylation status of candidate markers in plasma cfDNA. This multi-marker system demonstrates superior accuracy compared with individual markers and holds promising potential in clinical applications. This study also acknowledges certain limitations. Firstly, the relatively small sample sizes of samples derived from colorectal cancer patient tissue, blood leukocytes, and cfDNA have reduced the statistical power of our research findings. Secondly, a larger prospective validation is required, as this study only included patients from a single center, most of whom were identified due to presenting relevant symptoms, which might overestimate the diagnostic performance of the biomarkers in the real clinical world. Furthermore, the majority of non-cancer controls included in this study had not undergone colonoscopy to rule out polyps or adenomas, potentially introducing confounding factors and resulting in elevated false positives during the diagnostic performance evaluation. In our forthcoming endeavors, we aim to concentrate on conducting a comprehensive cfDNA sample analysis from multi-center patient cohorts, delving deeper into the identification of optimal cfDNA methylation biomarkers. By integrating these findings with other multi-omics markers, we aim to enhance the precision of colorectal cancer (CRC) diagnosis.

## 5. Conclusions

In summary, our analysis of whole-genome bisulfite sequencing across multiple sample sources of colorectal cancer patients has revealed a general hypomethylation pattern in colorectal cancer tumor tissues compared with normal tissues. We have identified a set of potential cfDNA methylation biomarkers for CRC diagnosis. Furthermore, we validated the disease relevance of three selected DMRs (*DAB1*, *FAM19A5*, and *PPP2R5C*), through methylation diagnostic tests based on blood cfDNA. Our research findings provide new insights into the role of DNA methylation in the development of colorectal cancer and demonstrate the utility of cfDNA methylation biomarkers in CRC diagnosis. These findings also offer potential candidates for epigenetic analysis in CRC diagnostics.

## Figures and Tables

**Figure 1 biomolecules-14-00996-f001:**
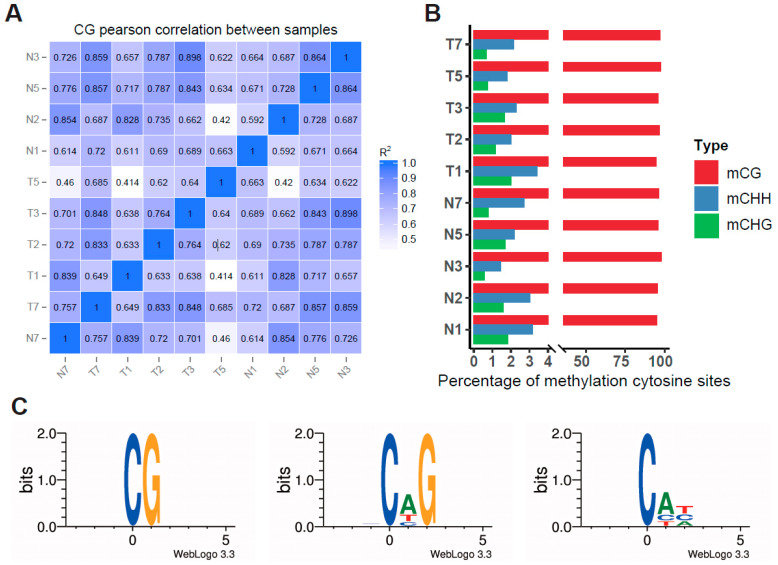
The DNA methylation characteristics of tumor tissues and adjacent tissues in colorectal cancer.(**A**) The correlation analysis of the methylation between samples. Heat maps show the correlation coefficient between samples; R^2^: the square of Pearson’s correlation coefficient. (**B**) Comparison of DNA methylation patterns in different samples. (**C**) Logo plots of the sequences proximal to mCG, mCHG, and mCHH sites.

**Figure 2 biomolecules-14-00996-f002:**
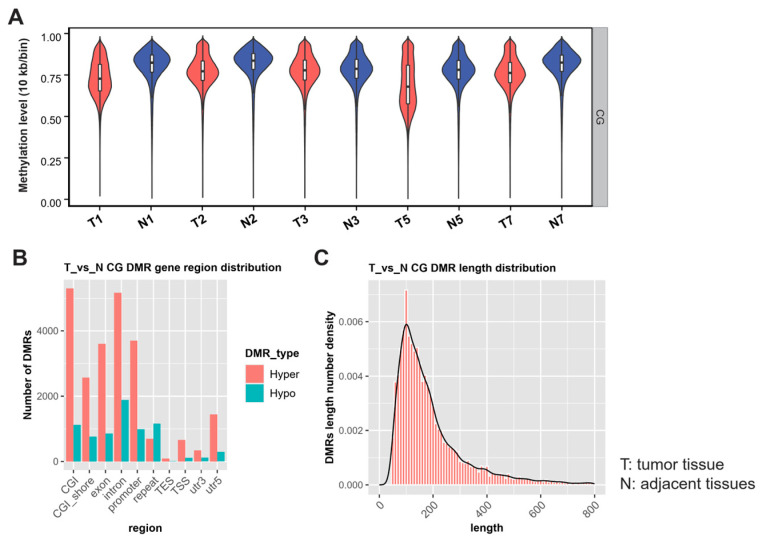
Characteristics of the DMRs. (**A**) The overall genome methylation level of tumor tissues was significantly lower than that of adjacent tissues(N) (*p* < 0.05). (**B**) The distribution of DMRs in different functional regions. (**C**) The frequency distribution of DMR counts in length.

**Figure 3 biomolecules-14-00996-f003:**
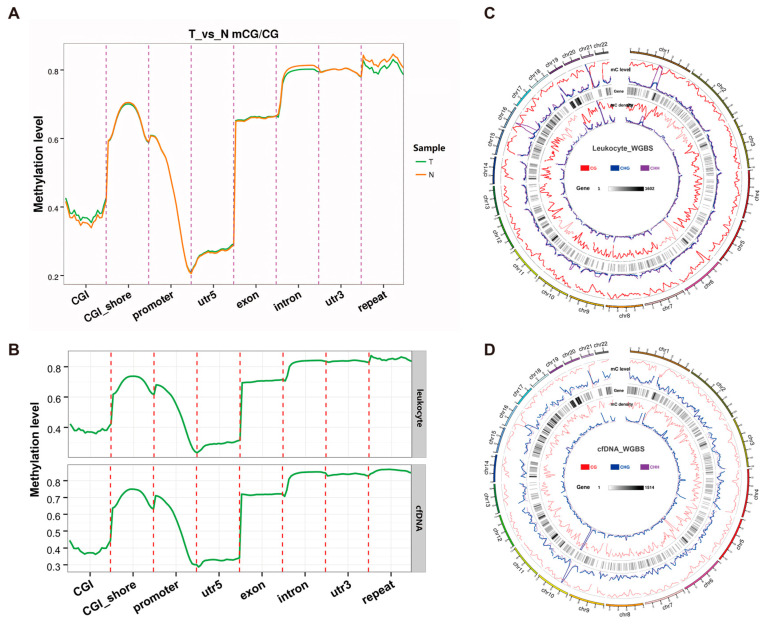
Comparison of methylation levels and distribution among different samples. (**A**) Global DNA methylation levels in different functional regions between tumor tissues and adjacent tissues. (**B**) Global DNA methylation levels in different functional regions between peripheral blood leukocytes and cfDNA. (**C**) Circos map of chromosome methylation levels in peripheral blood leukocyte samples. (**D**) Circos map of chromosome methylation levels in cfDNA samples. Note: T, tumor tissue; N, adjacent normal tissue. Circos map: from the outside to the inside, the methylation level is displayed linearly, while the gene-number density heat map is displayed linearly. Internal scale: three sequence environments (CG in red; CHG in blue; CHH in purple), gene density heat scale: the scale from gray to black indicates the number of genes from low to high.

**Figure 4 biomolecules-14-00996-f004:**
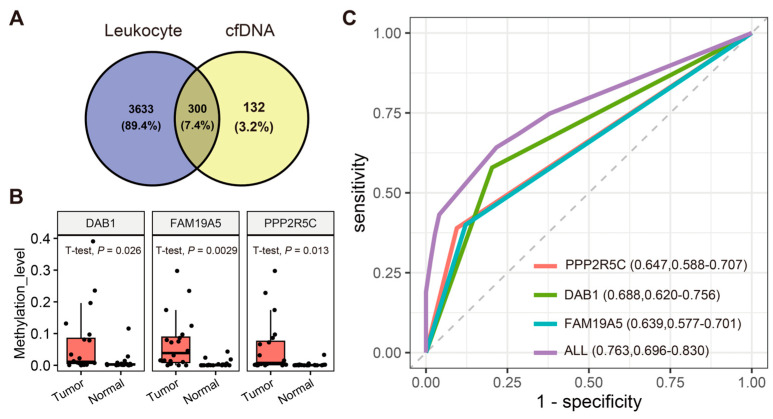
Detection of colorectal cancer using DMR-dependent cfDNA fragmentation profiles. (**A**) Venn diagram-based selection of DMRs detected only in cfDNA with higher methylation levels in tumor tissue. (**B**) Comparison of methylation levels between matched tumor tissue and adjacent normal tissue samples. (**C**) ROC curve for three DMRs in the *DAB1*, *PPP2R5C*, and *FAM19A5* genes and their combined analysis.

**Table 1 biomolecules-14-00996-t001:** The summary of data generated by genome-wide bisulfite sequencing.

Sample ID	Raw Reads	Clean Reads	Clean Rate (%)	Unique Mapping Rate (%)	Q20 (%)	Q30 (%)	BS Conversion Rate (%)
T1	400000000	390218278	89.18	81.85	96.36	90.53	99.733
T2	363290335	354589208	89.2	81.58	96.32	90.7	99.711
T3	383233529	373609013	89.08	77.33	96.85	90.2	99.72
T5	400000000	390459772	89.18	81.74	96.94	90.64	99.686
T7	413942035	406535734	89.6	84.12	96.7	89.25	99.512
N1	357706480	348563492	89.18	78.87	96.91	90.35	99.704
N2	390501803	380479444	89.09	80.17	96.79	90.48	99.708
N3	360000000	351523494	89	78.91	96.84	89.76	99.684
N5	400000000	391234005	89.38	80.14	96.36	89.48	99.711
N7	410933027	403303232	89.59	83.11	96.5	89.36	99.522
Leukocyte	306958167	297417716	86.83	81.4	97.4	93.04	99.641
cfDNA	75294759	72532206	85.83	78.46	97.77	92.97	99.056

## Data Availability

The datasets generated and/or analyzed during the current study are available from the corresponding author upon reasonable request.

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
