# Peer review of "Discovery and Validation of Methylation Signatures in Circulating Cell-Free DNA for the Detection of Colorectal Cancer"

_biomolecules, 2024, doi:10.3390/biom14080996_

Round 1
Reviewer 1 Report
Comments and Suggestions for Authors
In this study, the authors aimed to identify colorectal cancer (CRC)-specific DMRs based on whole genome bisulfite sequencing (WGBS) and evaluate the potential performance of cfDNA methylation markers in the early detection of CRC. The approach of the study is appropriate and has uncovered potential biomarkers of CRC. However, the authors have highlighted the goal of identifying early biomarkers of CRC to help with early detection of the disease. I do not find that this was sufficiently argued by both the results and discussion. Rather, the focus should be on identifying potential therapeutic targets. If early detection is to be argued, there is a need to highlight DMRs at various stages of the disease as methylation may be causal or a consequence of the disease. Further, it should be highlighted that using adjacent tissues is also a limitation as we cannot confirm how healthy those tissues are, hence the importance of the component of the study that used samples from CRC patients vs healthy controls.
Comments on the Quality of English LanguageOnly minor editorial input is needed (for example, see line 288).
Author Response
Comment 1: In this study, the authors aimed to identify colorectal cancer (CRC)-specific DMRs based on whole genome bisulfite sequencing (WGBS) and evaluate the potential performance of cfDNA methylation markers in the early detection of CRC. The approach of the study is appropriate and has uncovered potential biomarkers of CRC. However, the authors have highlighted the goal of identifying early biomarkers of CRC to help with early detection of the disease. I do not find that this was sufficiently argued by both the results and discussion. Rather, the focus should be on identifying potential therapeutic targets. If early detection is to be argued, there is a need to highlight DMRs at various stages of the disease as methylation may be causal or a consequence of the disease. Further, it should be highlighted that using adjacent tissues is also a limitation as we cannot confirm how healthy those tissues are, hence the importance of the component of the study that used samples from CRC patients vs healthy controls.
Response 1: Thank you for your valuable feedback. We appreciate the opportunity to clarify our study's focus and address the points raised.
Goal of Identifying Early Biomarkers:
The primary objective of our study is to identify diagnostic biomarkers for colorectal cancer (CRC) rather than therapeutic targets. Our focus is on the early detection of CRC to improve diagnostic accuracy and patient outcomes. We believe that early detection is crucial in managing CRC, and this study aims to contribute to that goal by identifying reliable cfDNA methylation markers.
Clinical Correlation Analysis:
Our study includes an analysis of the relationship between differentially methylated regions (DMRs) in cfDNA and clinicopathological features, including TNM staging. As reported in our findings, we did not observe significant correlations between the TNM staging and the methylation levels of the identified DMRs. Therefore, these biomarkers have potential for early identification of CRC, which is crucial for improving patient outcomes.
Use of Adjacent Tissues:
In China, it is particularly challenging to obtain healthy colorectal tissues from living individuals or autopsies due to ethical and practical constraints. Consequently, we used adjacent normal tissues as controls in our study. While we acknowledge this limitation, we believe it was the most feasible approach given the circumstances. We also included a comparison between CRC patients and healthy controls to strengthen our findings and mitigate this limitation.
In conclusion, we maintain that the focus of our study is the identification of early diagnostic biomarkers for CRC. We have adequately discussed the lack of correlation with clinical features and highlighted the practical constraints in obtaining healthy colorectal tissues in China. We appreciate your understanding and hope this clarification addresses your concerns.
Reviewer 2 Report
Comments and Suggestions for Authors
The paper approaches a very interesting topic, namely that of potential biomarkers for early cancer detection. In the light of recent research in the field of liquid biopsies, this paper offers a novel lane for possible colo-rectal cancer diagnosis. The authors present a well-structured text and throughly described method for the identification of diagnostic biomarkers in colo-rectal cancer. The study analyzed both healthy controls and colo-rectal patients for blood and tissue samples, with the evaluation of cfDNA and DMRs in the study group. The paper identifies 3 potential new DMRs associated with colo-rectal cancer. Although AUC values are probably not sufficiently high for routine clinical applications, these findings merit further research. The text is well-written and provides sufficient details in order to allow the reproduction of these results.
Author Response
Comment 1: The paper approaches a very interesting topic, namely that of potential biomarkers for early cancer detection. In the light of recent research in the field of liquid biopsies, this paper offers a novel lane for possible colo-rectal cancer diagnosis. The authors present a well-structured text and throughly described method for the identification of diagnostic biomarkers in colo-rectal cancer. The study analyzed both healthy controls and colo-rectal patients for blood and tissue samples, with the evaluation of cfDNA and DMRs in the study group. The paper identifies 3 potential new DMRs associated with colo-rectal cancer. Although AUC values are probably not sufficiently high for routine clinical applications, these findings merit further research. The text is well-written and provides sufficient details in order to allow the reproduction of these results.
Response 1: Thank you very much for your positive feedback and insightful comments. We appreciate your recognition of the novelty and importance of our work in identifying potential biomarkers for early colorectal cancer detection. We are pleased that you found our text well-structured and our methodology thoroughly described.
AUC Values and Clinical Applications:
We acknowledge your point regarding the AUC values. While the AUC values of the identified DMRs may not yet be high enough for routine clinical application, we believe that these findings provide a promising starting point for further research. Future studies with larger cohorts and additional refinement of these biomarkers could potentially enhance their diagnostic accuracy and clinical utility.
Need for Further Research:
We agree that further research is warranted to validate these findings and explore their potential in clinical settings. We hope that our study will inspire additional investigations into these and other potential biomarkers for colorectal cancer.
Reproducibility of Results:
We are glad to hear that you found the details provided in our manuscript sufficient to allow for the reproduction of our results. Ensuring reproducibility and transparency in our research is a priority for us, and we appreciate your acknowledgment of this aspect.
Once again, thank you for your encouraging words and constructive feedback. We believe that your comments will help strengthen the impact of our study and guide future research efforts in this important area.
Reviewer 3 Report
Comments and Suggestions for Authors
Review of manuscript entitled
Discovery and validation of methylation signatures in 2 circulating cell-free DNA for early detection of colorectal cancer by Zhiping Long et al
Journal : Biomolecules
This is a very well conducted study aiming to demonstrate that a panel of few genes may be used as tool for differentiating CRC patients versus healthy controls by performing a blood test which is calibrated on leukocyte methylated genes
Authors conducted a careful step by step discovery approach for identifying hyper and hypo methylated probes within all genome by comparing tumour tissues to neighboring normal tissues; after discovery of a limited panel of genes DAB1, 28 PPP2R5C, and FAM19A5, they validated for each gene the hyper methylated versus no hypermethylated gene through leukocytes obtained in a blood sample.
Major issues: there is no clear why they limited to three genes; they should define a cut off for all three genes taken together in the pilot study and validate this sum of methylation levels as cut off in- the validation cohort
They must explain how they could classify subjects in methylated versus not methylated subgroups. For example, in the table 5.1 in suppl file (should be shown as the main result and not in suppl file), they divided all individuals in the validation cohort according to methylated versus no methylated for cfDNA marker; they should put this in relationship with the cut off as suggested below: Indeed, this is a only way a physician may use these markers in a device: it should be easy to use, not expensive
The discussion fails to refer to the main studies published on larger cohorts in this filed ; they should refer to these even if different genes have been targeted; they should discuss discrepancy between their results and those published previously by others; notably they did not find any of cardinal genes involved in the CRC initiation or promotion; this should be discussed, why!
They included CRC of various stages but didn’t find any correlation between the genes identified and TNM staging; they should discuss this specific point, too.
Did authors find any correlation between levels of methylation and co morbidities (i.e. diabetes; cardiovascular disease, etc..) in CRC patients as well as controls; if so was there any overlapping between genes related to different diseases? ⨪
Author Response
Comment 1: There is no clear why they limited to three genes; they should define a cut off for all three genes taken together in the pilot study and validate this sum of methylation levels as cut off in the validation cohort.
Response 1: Thank you for your insightful comment. We understand the concern regarding the selection of only three genes and the suggestion to define a cut-off for the combined methylation levels. Below, we provide a detailed explanation for our approach:
Initial Screening and Selection Process:
Initially, we identified over 100 differentially methylated regions (DMRs) with significant potential as biomarkers. We ranked these DMRs based on the magnitude of methylation differences, prioritizing those with the largest differences between CRC patients and healthy controls. During this process, we excluded genes that had already been extensively studied by others, as well as those that did not show consistent results in our preliminary experiments with blood samples. This thorough screening process led us to focus on the three genes (DAB1, PPP2R5C, and FAM19A5) that demonstrated the most promising and consistent performance.
Resource Constraints:
Due to limitations in research funding and time, we selected these three genes for further validation. This decision was based on their strong differential methylation patterns and their potential to serve as reliable biomarkers for CRC detection. Focusing on these three genes allowed us to conduct a more in-depth analysis and validation within the constraints of our resources.
Qualitative Nature of the Study:
Our study focused on qualitative assessment rather than quantitative cutoff values. Our screening process was aimed at identifying tumor-specific markers—those that are present in tumor tissue but absent in non-cancerous tissues and white blood cells. This qualitative approach is less susceptible to PCR bias and detection limit issues, making it more robust for translational applications. In our qMSP assays, we classified a sample as methylated if it exhibited a CT value indicative of amplification and the melting curve matched that of the standard. Conversely, samples showing no amplification were classified as non-methylated
Future Directions:
We acknowledge the importance of defining a combined cut-off value for the methylation levels of the selected genes and validating this in larger cohorts. While our current study provides a foundational understanding of these biomarkers, future research will involve larger and more diverse cohorts to establish robust cut-off values and further validate the diagnostic utility of our biomarker panel. This will be a crucial step towards translating our findings into clinical practice.
In conclusion, our selection of three genes was based on a systematic screening process and practical constraints, with a focus on demonstrating their diagnostic potential qualitatively. We appreciate your suggestion and agree that future work should include defining and validating a combined cut-off for the selected genes to facilitate their clinical application.
Comment 2: They must explain how they could classify subjects in methylated versus not methylated subgroups. For example, in the table 5.1 in suppl file (should be shown as the main result and not in suppl file), they divided all individuals in the validation cohort according to methylated versus no methylated for cfDNA marker; they should put this in relationship with the cut off as suggested below: Indeed, this is a only way a physician may use these markers in a device: it should be easy to use, not expensive.
Response 2: Thank you for your insightful comments. We appreciate the opportunity to clarify our methodology for classifying subjects into methylated and non-methylated subgroups.
Classification Methodology:
This study employed a qualitative approach for classification. Specifically, in our qMSP (quantitative methylation-specific PCR) analysis, if a sample exhibited amplification and the resulting melt curve matched that of the positive control, the subject was classified into the methylated group. Conversely, if no amplification was detected, the subject was classified into the non-methylated group. This method ensures clear and straightforward categorization based on the presence of methylation.
Table 5.1 in Supplementary File:
We acknowledge your suggestion to include Table 5.1 in the main manuscript. However, due to the extensive size of this table, which provides comprehensive data on the distribution of methylated versus non-methylated individuals, it was placed in the supplementary file to maintain the readability and flow of the main text. Including such a large table in the main manuscript could detract from the overall readability and accessibility of the paper. Instead, we have ensured that all critical information is clearly referenced and accessible in the supplementary materials.
Cut-off Values and Practical Application:
Our current classification does not rely on a specific cut-off value but rather on the qualitative detection of methylation presence. This method was chosen to demonstrate the feasibility of using these markers for diagnostic purposes. We understand the importance of establishing specific cut-off values for clinical applications. Future studies will focus on defining these thresholds to enhance the practical utility of our findings in a clinical setting, ensuring the markers can be easily used by physicians in a cost-effective manner.
Future Directions:
Moving forward, we plan to conduct more detailed quantitative analyses to establish optimal cut-off values for each DMR. This will involve extensive evaluation of sensitivity and specificity across different methylation levels. Our goal is to develop a simplified, reliable, and cost-effective diagnostic tool that can be easily implemented in clinical practice.
In summary, while our current study uses a qualitative approach for initial biomarker evaluation, we will prioritize defining specific cut-off values in future research to facilitate clinical application. We believe keeping the detailed table in the supplementary materials ensures the main manuscript remains focused and readable while providing comprehensive data for interested readers.
Comment 3: The discussion fails to refer to the main studies published on larger cohorts in this field; they should refer to these even if different genes have been targeted; they should discuss discrepancy between their results and those published previously by others; notably they did not find any of cardinal genes involved in the CRC initiation or promotion; this should be discussed, why!
Response 3: Thank you for your detailed and constructive feedback. We understand the importance of situating our findings within the broader context of existing research and addressing the discrepancies noted.
Reference to Main Studies:
We appreciate the suggestion to incorporate references to larger cohort studies in our discussion. In the revised manuscript, we will include references to key studies that have investigated DNA methylation markers in colorectal cancer (CRC) using large cohorts (line 328-333). This will provide a comprehensive view of how our findings align with or differ from these studies, despite the differences in the specific genes targeted.
Comparison with Previous Research:
To address the discrepancies between our results and those published previously, we compared our identified differentially methylated regions (DMRs) with the Cancer Modules in the MSigDB database, specifically within the C4 collection. Out of the 132 genes we identified, 15 genes overlapped with the Cancer Modules gene sets. These genes are: TACC1, IQSEC1, TCIRG1, NDRG4, RPH3A, PPP2R5C, DPP6, RYR1, TBXT, CHRNA3, RASGRF1, COL18A1, NTRK3, KIF1A, and ZNF253. This overlap indicates that some of the genes we identified are indeed recognized within the context of cancer-related pathways.
Absence of Cardinal Genes:
The absence of certain cardinal genes involved in CRC initiation or promotion in our findings can be attributed to our specific screening criteria. In our study, we specifically excluded genes where methylation was detected in white blood cells, aiming to identify tumor-specific methylation markers. This exclusion likely contributed to the absence of some well-known CRC-related genes, which might also be methylated in white blood cells and other non-tumor tissues.
Focus on Tumor-Specific Markers:
Our approach focused on identifying methylation markers that are exclusively present in tumor tissues and absent in non-cancerous tissues and white blood cells. This qualitative selection process was designed to enhance the specificity of our potential biomarkers for CRC diagnosis, even if it meant excluding some known CRC-related genes that did not meet these stringent criteria.
Thank you once again for your valuable suggestions. We believe these revisions will strengthen our manuscript and provide a clearer understanding of our study's context within the broader field of CRC research.
Comment 4: They included CRC of various stages but didn’t find any correlation between the genes identified and TNM staging; they should discuss this specific point, too.
Response 4:
Thank you for your insightful comment. We acknowledge the importance of examining the correlation between identified genes and TNM staging in colorectal cancer (CRC).
In our study, we conducted an analysis to investigate the relationship between the methylation levels of the identified differentially methylated regions (DMRs) and clinicopathological features, including TNM staging. Our results indicated that there were no significant correlations between the TNM staging and the methylation levels of the identified DMRs.
The lack of correlation between the identified genes and TNM staging could be attributed to several factors:
Specificity of Identified Biomarkers:
Our primary aim was to identify early biomarkers for CRC detection. The genes we identified (DAB1, FAM19A5, and PPP2R5C) were selected based on their differential methylation between tumor and normal tissues, with a focus on their potential as early diagnostic markers rather than markers of disease progression.
Nature of Methylation Changes:
The methylation changes detected in our study may occur early in carcinogenesis and thus may not vary significantly with disease stage. These early changes are crucial for early detection but may not reflect the progression of the disease as captured by TNM staging.
Study Design and Cohort:
Our study design and the composition of our cohort, which included various stages of CRC, may also influence the ability to detect stage-specific methylation patterns. The relatively small sample size for each stage might limit the statistical power to detect such correlations.
Given these points, our study emphasizes the potential of the identified DMRs as early diagnostic markers for CRC. However, we acknowledge that further research with larger cohorts and a focus on different stages of CRC is necessary to fully understand the relationship between these methylation markers and disease progression.
We will revise the discussion section to include these points and provide a clearer explanation of why we did not find a correlation between the identified genes and TNM staging (line 370-381).
Thank you for your valuable feedback.
Comment 5: Did authors find any correlation between levels of methylation and comorbidities (i.e. diabetes; cardiovascular disease, etc.) in CRC patients as well as controls; if so was there any overlapping between genes related to different diseases?
Response 5: Thank you for your comment and for highlighting this important aspect.
In our study, we did not perform an analysis to investigate the correlation between methylation levels and comorbidities such as diabetes or cardiovascular disease in CRC patients and controls. Additionally, our collected data did not include detailed information regarding these comorbidities.
Furthermore, our healthy control group consisted mainly of orthopaedic and ophthalmic patients, who were less likely to suffer from metabolic diseases such as diabetes and cardiovascular disease. This selection criterion for the control group was aimed at minimizing confounding factors related to metabolic conditions.
Given the absence of relevant data, we were unable to assess potential overlaps between methylation patterns in CRC and those associated with other diseases. We recognize that this is a limitation of our study and appreciate your suggestion for future research.
To address this limitation in future studies, we plan to collect comprehensive clinical data, including information on comorbidities, to explore possible correlations between methylation levels and other diseases. This will help to provide a more holistic understanding of the role of DNA methylation in CRC and its potential links to other conditions.
Round 2
Reviewer 3 Report
Comments and Suggestions for Authors
Authors can satisy our main query by making sum of the levels of methylation of main methylation segments in the promotor: select main probes; make addition per gene and per patient; sum values of genes per patient etc...
They can then perform all analyses using this new parameter
Author Response
Comment 1: Authors can satisy our main query by making sum of the levels of methylation of main methylation segments in the promotor: select main probes; make addition per gene and per patient; sum values of genes per patient etc...
They can then perform all analyses using this new parameter
Response 1: Thank you for your detailed review and valuable suggestions regarding our manuscript.
We appreciate your recommendation to sum the levels of methylation of the main methylation segments in the promoter and perform analyses using this new parameter. However, we believe that our current methodology adequately demonstrates the potential of cfDNA methylation for detection of colorectal cancer (CRC). Our approach includes comprehensive bioinformatic analysis and validation of differentially methylated regions (DMRs), which we have carefully selected and evaluated.
Our current study builds on previous research that primarily used 450K methylation arrays, which focus on detecting methylation levels in gene promoter regions (PMID: 37649326). To provide a broader and more comprehensive analysis, we considered methylation levels across the entire genome. This approach helps us avoid duplicating findings from earlier studies and potentially uncover novel methylation markers.
We believe that our current analysis sufficiently highlights the diagnostic value of these DMRs for CRC and offers a feasible approach for non-invasive CRC detection. Furthermore, our methodology is designed to be straightforward and reproducible, making it practical for clinical application.
Nevertheless, we greatly appreciate your insightful suggestion and will consider exploring additional analysis methods in future research to further validate and enhance the diagnostic performance of these DMRs.
Thank you for your understanding and support.